# The role of telemedicine towards improved sustainability in healthcare and societal productivity in Turkey

Figen Özen[1], Alptug H. Kaynar[2], A. Kubilay Korkut[3], Melike Elif Teker Açıkel[4], Z. Dilsun Kaynar[5]*, A. Murat Kaynar[6,7,8,9]

**1** Electrical and Electronics Engineering Department, Haliç University, Eyüp, Istanbul, Turkey, **2** Columbia University, New York, NY, United States of America, **3** Department of Cardiothoracic Surgery, Haliç University, Eyüp, Istanbul, Turkey, **4** Department of Cardiothoracic Surgery, S.B.Ü. Haseki Eğitim ve Araştırma Hastanesi, Sultangazi, Istanbul, Turkey, **5** Computer Science Department, School of Computer Science, Carnegie Mellon University, Pittsburgh, PA, United States of America, **6** Department of Anesthesiology and Perioperative Medicine, University of Pittsburgh, Pittsburgh, PA, United States of America, **7** The Center for Innovation in Pain Care (CIPC), University of Pittsburgh, Pittsburgh, PA, United States of America, **8** Department of Critical Care Medicine, University of Pittsburgh, Pittsburgh, PA, United States of America, **9** The Clinical Research, Investigation, and Systems Modeling of Acute Illness (CRISMA) Center, University of Pittsburgh, Pittsburgh, PA, United States of America

* dilsunk@cmu.edu

**Citation:** Özen F, Kaynar AH, Korkut AK, Teker Açıkel ME, Kaynar ZD, Kaynar AM (2024) The role of telemedicine towards improved sustainability in healthcare and societal productivity in Turkey. PLoS ONE 19(12): e0314986. https://doi.org/10.1371/journal.pone.0314986

**Data Availability Statement:** Data cannot be shared publicly because of Turkish Privacy Laws. Anonymized data are available from the Halic University Ethics Committee for researchers who

## Abstract

The healthcare systems of low and middle-income countries suffer from lack of resources that could be remedied by employing novel care strategies such as telemedicine [1]. Here, the hypothetical impact of delivering telemedicine care on environment and society in three busy cardio-vascular clinics in Istanbul, Turkey, is examined. The study exploits demographics, wages, productivity, and patient-specific data to develop a hypothetical telemedicine framework for the Turkish healthcare landscape. Specifically, the distance traveled and travel time to receive care using location of the clinics and patients addresses seeking care are tabulated. Data from August 3, 2015, to January 25, 2023 involves 45,602 unique encounters with 448 unique diagnoses recorded for the patient encounters, where the patients in the top 5% of the most common diagnoses traveled 23.82 ± 96.3 km to reach the clinics. Based on our model, telemedicine care for cardiovascular diseases would have saved 656,258 km if all patients were to take the first visit in person followed by telemedicine visits in lieu of face-to-face care for all visits. The travel-associated carbon footprint and wage losses for in-person care is calculated and exploiting telemedicine could have saved approximately 30% carbon footprint and prevented approximately $503,752.8 wage loss. It is possible that telemedicine could ease the burden on patients, environment, increase access, and prevent the wage losses caused by unnecessary hospital visits.

## 1. Introduction

Cardiovascular diseases (CVDs) are the leading cause of mortality in the world according to the recent Global Burden of Disease Study [1–3]. CVDs substantially contribute to loss of

meet the criteria for access to confidential data at the following address: Haliç University, 5. Levent Mahallesi, 15 Temmuz Şehitler Caddesi, No: 14/12 34060 Eyüpsultan/İSTANBUL, TURKEY.

**Funding:** The author(s) received no specific funding for this work.

**Competing interests:** The authors have declared that no competing interests exist.

health and excess health system costs [2–4]. Deaths from CVD surged almost by 60% globally over the last 30 years, increasing from 12.1 million in 1990 to 20.5 million in 2021 [2]. In 2021, out of the 20.5 million CVD-related deaths, almost 80% of the mortality took place in low- and middle-income countries (LMIC) [2].

Turkey, an upper middle-income country according to the most recent classification of the World Bank, is experiencing an increasing burden of noncommunicable diseases (NCD), which account for 86% of total deaths, and nearly 1 in 5 adults die prematurely [5–7]. CVDs are responsible for nearly half (47%) of all deaths in Turkey, where mortality is projected to increase by 2.3-fold in men and 1.8-fold in women by 2030. Turkey is among the countries with the highest CVD mortality in Europe and Turkish women have the highest overall mortality [8].

There is a cost to deliver and receive healthcare, including care for CVDs. With ever growing healthcare costs, governments are struggling to find ways to fulfill their promise of delivering healthcare to their citizens [9]. Rising healthcare costs are especially a growing concern in LMICs, including Turkey, where the cost of healthcare is predicted to rise from $378/person in 2019 to $694/person in 2050 [(https://www.healthdata.org/turkey), accessed 1 June 2024] [10, 11]. In addition to the ever-increasing strain on its limited health budget due to the steady growth of the aging population, it was reported that only 7.5 minutes were able to be allotted per doctor visit in hospitals run by the Ministry of Health in Turkey, worsening the strain on the system, providers, and patients [12]. Turkey's population growth and increase in healthcare cost are further compounded by the exodus of the healthcare workforce into other careers following the COVID-19 pandemic [13–18]. Thus, policy makers must come up with innovative care methods to help alleviate the cost to deliver and receive healthcare [19].

One such care method can be obtained by incorporating telemedicine, which allows providers to care for patients without an in-person office visit [20, 21]. Our study focuses on patients with CVDs and the potential savings afforded by telemedicine in travel distance and time, and lost wages. Delayed access to acute as well as chronic care is associated with increased mortality and prior work showed the negative impact of long wait periods on CVD outcomes [7–9]. We examined the feasibility of using telemedicine to follow-up non-urgent CVDs in three busy clinics in Istanbul, Turkey to lay the groundwork for a more efficient model system.

## 2. Materials and methods

### 2.1. Study design

The study was conducted in accordance with the Declaration of Helsinki. The study was approved by the Institutional Review Board of the Haliç University (5. Levent Mahallesi, 15 Temmuz Şehitler Caddesi, No: 14/12 34060 Eyüpsultan / İstanbul), Turkey (September 26, 2023; No. 212) and the research team accessed the data on October 15th, 2023 for research purposes. Consent was waived for our study.

We obtained datasets for demographics, healthcare costs, wages, productivity, and data of patients of the above-mentioned clinics to develop a hypothetical telemedicine framework. Specifically, we obtained the distance and time expended to receive care at the clinics by using the addresses of the clinics and patients seeking care. The calculated distances and travel times allowed us to calculate the carbon footprint and wage losses associated with traveling to seek in-person care in lieu of telemedicine appointments. We then calculated the savings in travel distance and time for hospital visits taking into account the cost of synchronous video conferences used for care.

The authors did not have access to patient names during or after data collection. The original dataset did include addresses associated with patient numbers. After the distance and time

data were obtained for each patient number, we further sanitized the data to exclude patient numbers and addresses. The sanitized dataset, which can be shared upon request, consists of a protocol number, medical diagnosis, distance traveled, and time spent associated with a given hospital visit for a patient. The range of protocol numbers has been modified to prevent indirect identification of patients using protocol numbers in the event that this dataset is used in conjunction with another dataset in which actual protocol numbers are used.

## 2.2. Population and settings

We studied outpatient clinic visits to three healthcare facilities in Istanbul focusing on cardiovascular diseases (Haseki Merkez, Haseki Sultangazi, Haseki 29 Mayıs) between August 3, 2015, to January 25, 2020. Ambulatory adults were the source of population for the study. Patients' addresses were the source of data for the study and demographic characteristics and clinical diagnoses were the variables collected from the patients' medical records.

## 2.3. Measurement outcomes

**2.3.1. Dependent and independent variables.** The independent variables were the addresses of patients and their destination clinic addresses. The dependent variables were the travel distance, time, $CO_2$ emission, and economic impact of traveling.

**2.3.2. Travel distance and time.** Travel distance and time were obtained by web scraping Google Maps travel time and distance data. The patient address dataset and patient demographical data were obtained from three clinics (Haseki Merkez, Haseki Sultangazi, and Haseki 29 Mayıs) in Istanbul serving a large catchment area (Fig 1). Using Selenium browser

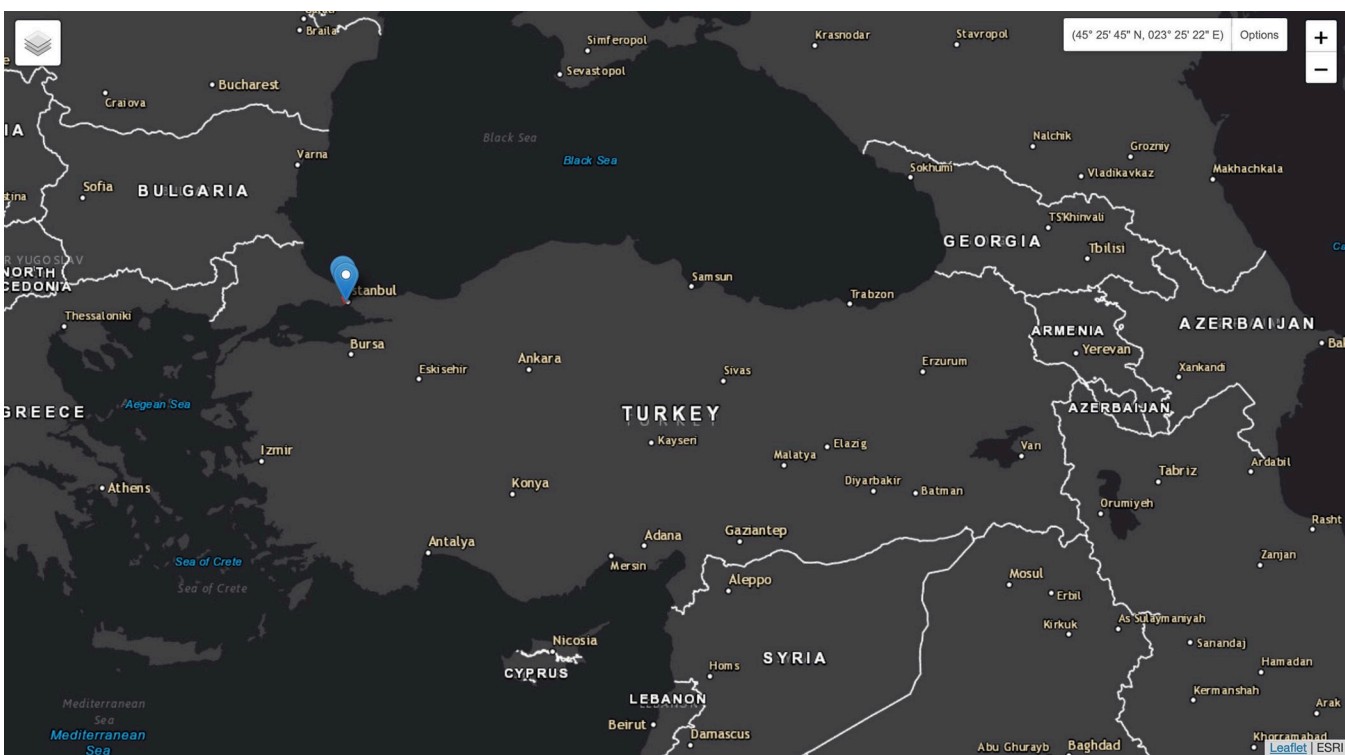

**Fig 1. Destination hospitals.** The patient address dataset and patient demographical data were obtained from three clinics (Haseki Merkez, Haseki Sultangazi, and Haseki 29 Mayıs) in Istanbul serving a large catchment area.

automation, each patient's address was matched to their respective branch of the Haseki hospital system in Istanbul, Turkey. This matched patient address-hospital pair was inserted into Google Maps and the resultant travel time and distance data, obtained on the "driving" setting, was recorded. The travel time is less reliable using the public transportation option as the times using public option are not predictable. The timing of the Google Map data extraction was performed after hours (5:00 PM until 9:00 AM EST, 9 AM until 5 PM UTC + 03:00) to minimize the impact of an inaccurate representation of traffic density on the time calculations [22].

**2.3.3. Carbon footprint of traveling to healthcare facilities and telemedicine alternatives.**  The carbon emissions were calculated based on assumed modes of transportation with models varying from personal vehicles to public transportation as well as the carbon emission from synchronous videoconference-based telemedicine. The carbon emission data per km is derived from the Global Change Data Lab [23] and the telemedicine carbon footprint data is obtained from the work by Obringer et al. [24]. In the example of in-person visit, a 10-minute doctor-patient interaction is assumed. Similarly, for telemedicine, the digital footprints are calculated for a 10-minute doctor-patient video conferencing, and it is assumed that on the average one large image is exchanged for diagnosis.

**2.3.4. Economic impact of traveling to healthcare facilities.**  Our dataset did not include any employment-related data. The lost wage cost is derived from the nation-specific wage and employment data from the International Labor Organization (ILO) for the year 2021 and the associated labor wage data for that year. The retirement age in Turkey is 58 for women and 60 for men. To estimate the number of people who were actually employed before that age, we used the employment rate data from the *Organisation for Economic Co-operation and Development* (OECD) [25, 26]. Two-way travel time and the additional estimated 10 min for physician interaction is multiplied by the hourly wage.

## 2.4. Data analysis

The diagnostic data for clinic visits were summarized and presented as the frequency. Descriptive data for distance, travel time, $CO_2$ emission, and wages as part of the economic impact were presented as total for individual diagnostic groups and mean ± for the whole cohort.

## 3. Results

### 3.1. The distance and time to travel to healthcare facilities

The dataset included 64,999 unique patient visits from August 3, 2015, to January 25, 2020, to the three healthcare facilities in Istanbul (Haseki Merkez, Haseki Sultangazi, Haseki 29 Mayıs). After correcting for missing or inaccurate address information, we had a complete address dataset of 45,602 unique patient encounters. There were 448 unique diagnoses recorded for the patient encounters and we decided to examine the top 5% of diagnoses (Fig 2).

Patients traveled on average 23.82 ± 96.3 km to reach the healthcare facilities (Figs 3 and 4). As shown in Fig 4, there would be a total of 656,258 km saved if all patients were to take the first visit in person followed by telemedicine for subsequent visits making it a potential saving. Our analysis assumes that the visits that follow the first in-person visit could be replaced with video conferencing. This is because our data showed that, for the diagnoses included in our analysis, 98.9% of the visits after the first were follow-up visits that did not require any in-person visits as per the expert panel.

# Top 5 Percent Diagnosis

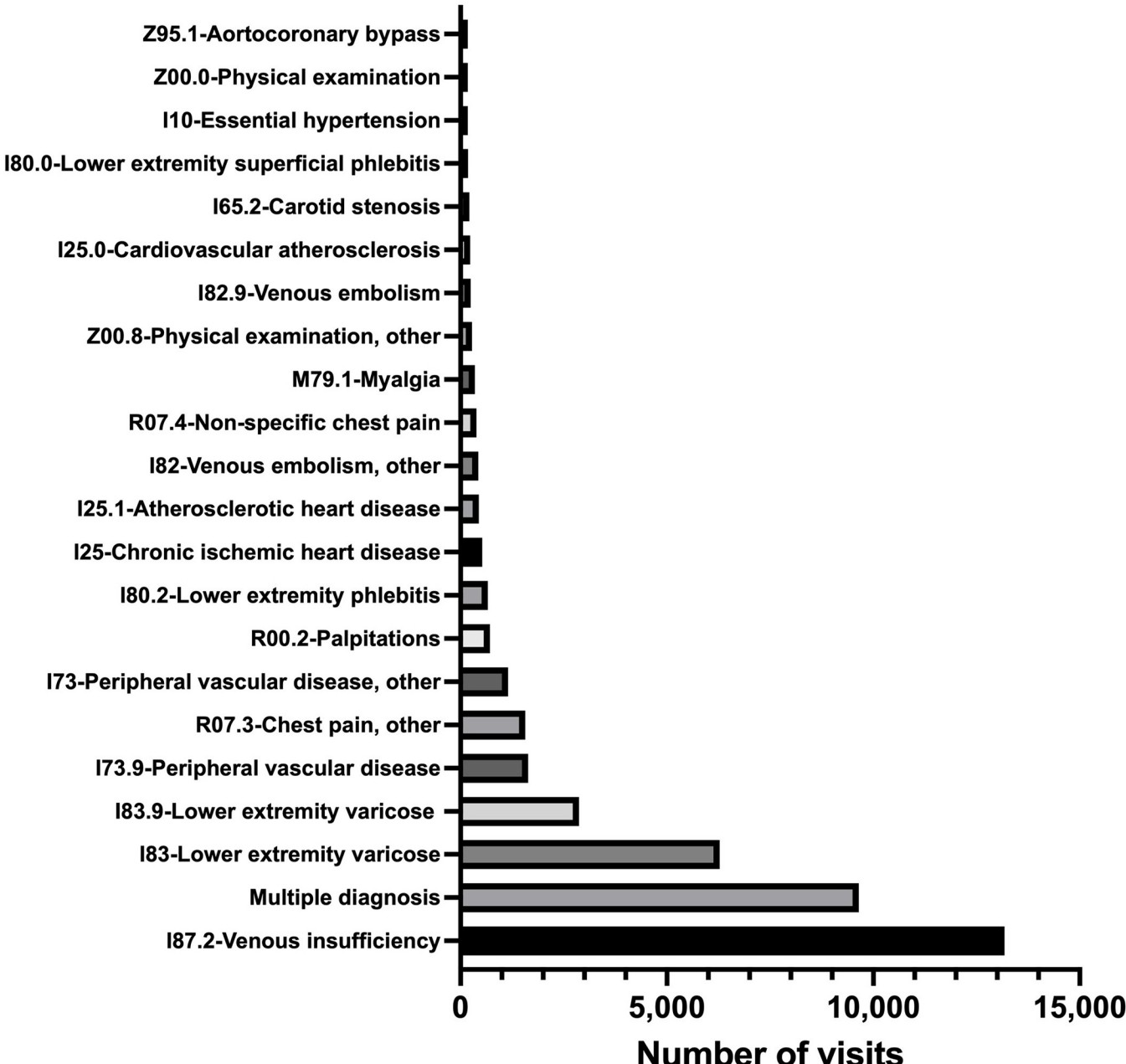

**Fig 2. The distribution of visit types.** The top 5% of diagnosis for the visits, mainly around cardiovascular diseases.

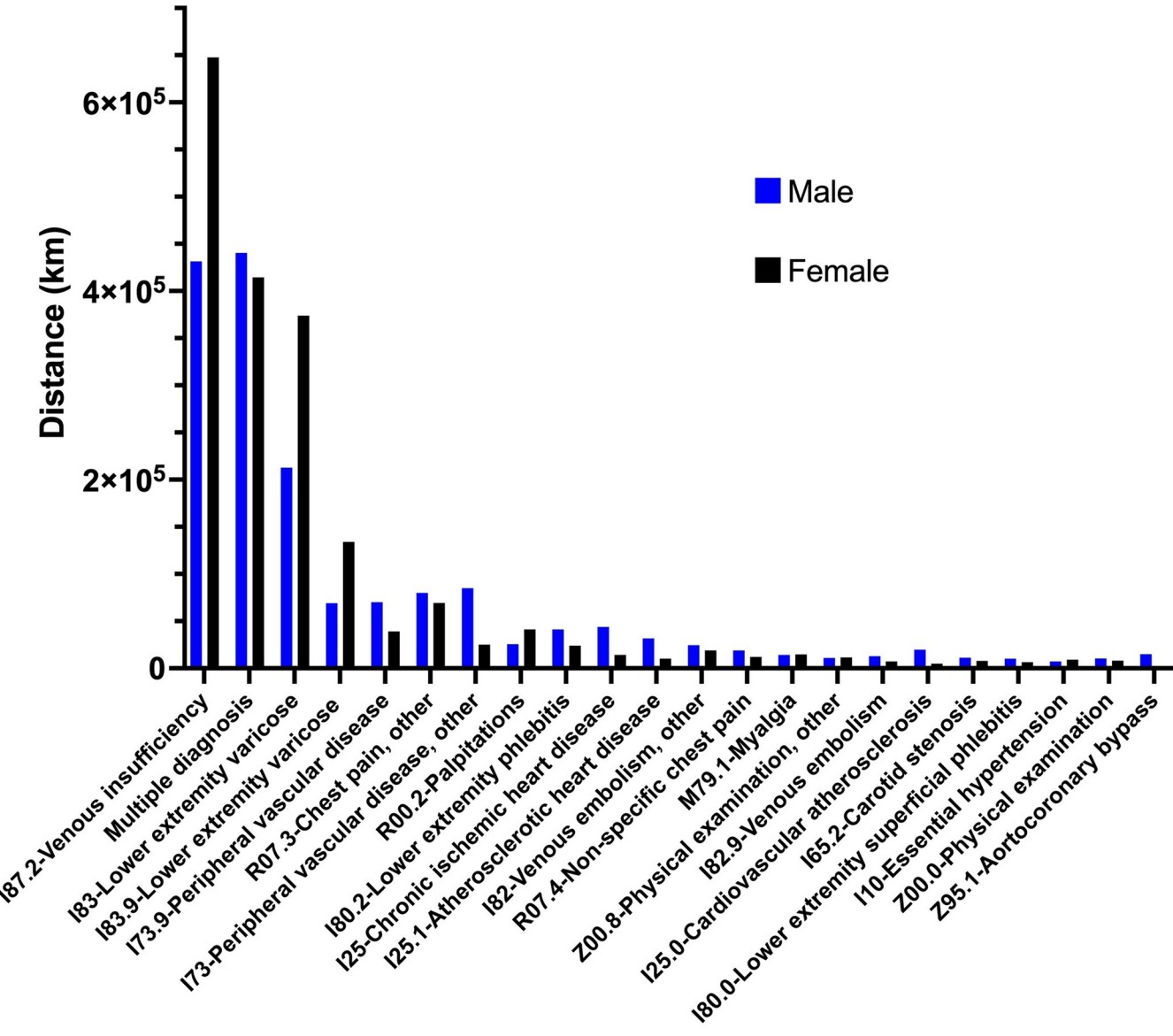

**Fig 3. Travel distance.** Distance traveled by the top 5% diagnosis groups.

## 3.2. Carbon footprint

In this study, calculations of carbon footprint were made to show the effects of hospital visits on the environment. Access to the hospital can be possible in various ways. However, to simplify the calculations, it is assumed that one can travel either by bus or car.

The carbon emissions were calculated based on assumed modes of transportation with models varying from personal vehicles to public transportation as well as the carbon emission from video-based telemedicine.

The carbon footprint of traveling by bus is 105 grams per kilometer and the carbon footprint of traveling by car is 192 grams per kilometer [27, 28]. Since it is necessary to make round-trip calculations in carbon footprints, the figure found was multiplied by two. For

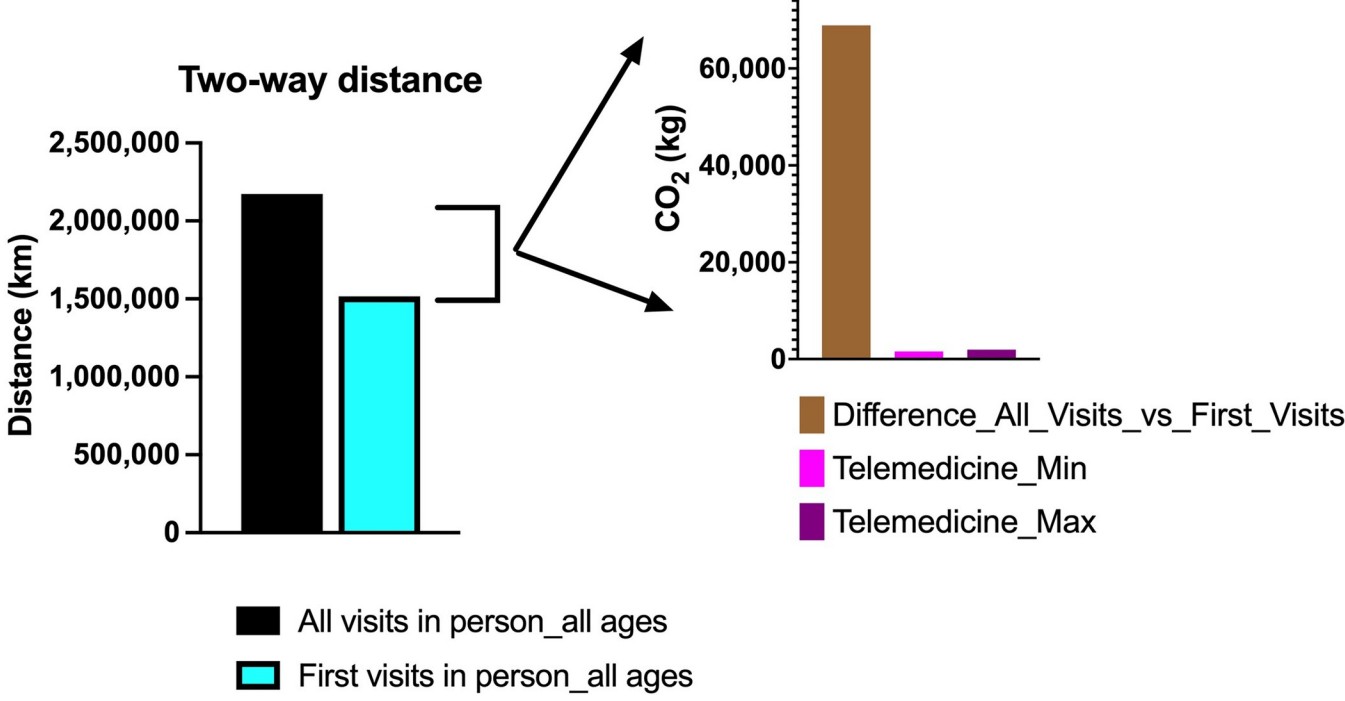

**Fig 4. Travel distance and $CO_2$ emission savings.** The distance and $CO_2$ difference if the first visits are done in person with subsequent televisits.

example, if the distance from the address of the patient to the hospital is 10 kilometers:

$$Carbon\ footprint\ by\ bus = 2*10*105 = 2100\ grams = 2.1\ kilograms$$

$$Carbon\ footprint\ by\ car = 2*10*192 = 3840\ grams = 3.84\ kilograms$$

From the above calculations, the cost of traveling by car is 82.86% higher than traveling by bus. On the other hand, if the patient in the example does not go to the hospital and is examined remotely, the footprint is minimal.

The digital footprints were calculated for a ten-minute doctor-patient video conferencing, and it is assumed that on the average one large image is exchanged for diagnosis. The digital footprint is independent of the distance between the hospital and the address of the patient. The calculations are as follows:

$$Minimum\ digital\ footprint = 2*2.53 + 2*50 = 105.06\ grams$$

$$Maximum\ digital\ footprint = 2*26.22 + 2*50 = 152.44\ grams$$

In min (max) digital footprint calculation, a ten-minute video conference cost is taken as 2.53 (26.22) grams and is multiplied by two to account for both the doctor and the patient. The added term is constant, and it is the cost of sending and receiving one large image file. It is assumed that the footprints of sending an image file and receiving it are approximately equal. The footprint values in the calculations except for the image are taken from Obringer et al. [24]. On the other hand, the footprint of an image is taken from Frost [29]. The dataset in this study contains 45,602 visits to the hospital and the total distance of the roundtrips by the patients turns out to be 2,173,262 kilometers. If all the roundtrips are made by bus, then the

## Two-way $CO_2$ footprint by bus

**Fig 5. $CO_2$ footprint by bus.** The difference for public transportation if the first visits are done in person with subsequent televisits.

total amount of carbon footprint is 228,192.61 kilograms for all in-person visits, whereas it could decrease to 159,285 kilograms if the first visits were done in person with subsequent visits via telemedicine (Fig 5).

Similarly, if all visits are made by car, then the total amount is 417,266.49 kilograms, which would decrease to 291,264 kilograms if the first visit were done in person with subsequent visits via telemedicine (Fig 6). The real amount is expected to be in the range of 159 to 417 tons.

Our assumption that all but first in-person visits could be replaced with telemedicine without compromising quality of patient care relies on the fact that, for the diagnoses that we selected, 98.9% of the visits subsequent to the first were follow-up visits that did not require any in-person intervention. This is also in line with the opinions of the cardio-vascular specialists working at these facilities we consulted while conducting our study.

## Two-way $CO_2$ footprint by car

**Fig 6. $CO_2$ footprint by car.** The difference for personal vehicle transportation if the first visits are done in person with subsequent televisits.

Table 1. Wage loss caused by in-person hospital visits.

| | Males < 60 years | Females < 58 years | Males (<60) and Females (<58), combined |
|---|---|---|---|
| Total visits | 13,995 | 19,656 | 33,651 |
| Travel times (hr) | 15,812.6 | 18,782.7 | 34,595.3 |
| All visit with added 10 min per visit for in-person travel times (hr) | 18,145.1 | 22,058.7 | 40,203.8 |
| Wage loss for an 8-hr day per visit, assuming employment at 57.4%, ($) | 209,504.1 | 294,248.7 | 503,752.8 |
| Wage loss per hr., assuming employment at 57.4%, ($) | 42,683.1 | 53,537.4 | 96,220.5 |

In this scenario, the total distance that must be traveled is 1,517,004.92 kilometers, which means a 30.20% decrease in the total amount of distance to be traveled. In this scenario, the number of visits is 29,985, in other words 34.25% less than the 'all visits in person' scenario. The effect on carbon footprint is also dramatic. If the first visits are done by bus and the rest via telemedicine, the difference of the carbon footprint is 68,907.10 kilograms. On the other hand, if the first visits are done by car, the difference of the carbon footprint is 126,001.55 kilograms. In both cases, a 30.20% reduction in the carbon footprint is achieved (Figs 4–6).

### 3.3. Economic impact

Wages were derived from the ILO site for the study period for Turkey and the average wage loss is calculated by multiplying the number of visits to the total time spent for traveling and visits. For the physician visits, a 10-min time is allotted. The 2021 average hourly wage for Turkey was 28.13 Turkish Lira (TL), where one USD was worth 8.86 TL in 2021, making the average hourly wage $3.26 [30]. As we did not have employment data for our cohort of patients, in order to model potential wage savings, we used the OECD data for employment rate. This data showed the average employment to be 57.4% between 2015 through 2022 for the 15–64 age range regardless of the gender overlapping with the study dataset.

Table 1 shows the travel times with an additional 10-min for in-person visit for working-age males and females in our cohort. The wage lost is presented based on the time spent traveling and in-person visit ("Wage loss per hour") as well as a whole day loss of productivity based on the number of visits. Additionally, these travel times are calculated based on the use of a personal vehicle, where the use of public transportation would increase the travel times.

## 4. Discussion

Our study focused on CVDs and the potential savings afforded by telemedicine in travel distance and time, $CO_2$ emissions, and lost wages as delayed access to acute as well as chronic care is associated with increased mortality [31–33].

The busy cardiology and cardio-vascular surgery clinics in Istanbul used in our study can also be viewed as a model for studying the potential positive patient outcomes, environmental, economic, and societal ramifications of deploying telemedicine. Moreover, the location of these clinics can demonstrate the benefits of telemedicine in settings where specialized care is concentrated in certain clinics in a populous city only accessible via often traffic-laden travel routes.

The findings of our study, based on the model we developed using the existing patient-level data, suggests (1) telemedicine-associated benefits to the environment, and (2) savings in wages.

Literature demonstrated similar outcomes in telemedicine treatment compared with face-to-face treatment for patients with peripheral arterial and venous diseases [34–37]. A recent work compared the use of telemedicine to in-person doctor visits among patients with

congestive heart failure (CHF). Among 850 patients with CHF, a hybrid telemedicine care model was not inferior to in-person visits for mortality [38]. Another trial showed decreased emergency medicine department admissions among patients with CHF if they were followed up by a tele-pharmacist [39]. Based on the clinical trials, there is an important and effective potential role for telemedicine in healthcare [40–42].

The global carbon footprint of healthcare was estimated to be 2 gigatons of $CO_2$-equivalents ($CO_2$e) in 2014, approximating 4.4% of global emissions, whereas global transportation contributed 7% of this total. Thus, in a world where climate change is a major risk for human health, the healthcare paradoxically continues to significantly contribute to this hazard [43, 44]. Along the environmental pressures of delivering healthcare, to deliver it in a sustainable fashion is a growing concern for every country because of both increasing costs and the increasing ratio of health expenditure to gross domestic product (GDP) [45]. For example, in Turkey, in the two decades until 2020, healthcare expenditure increased from $432/capita in 2000 to $1,305/capita in 2020 [46], nearly tripling in 20 years and the ratio of health expenditure to GDP increased from 4.46% to 5% [47, 48]. This is not an isolated phenomenon and between 2000 and 2020 as the worldwide health expenditure to GDP increased from 8.62% to 10.89% [49].

In addition to increasing healthcare cost, each visit to a healthcare facility results in increased carbon emissions due to travel and loss of productivity -unless the visit leads to gain in health and subsequent productivity [50–55]. The travel-related carbon emission within the context of global warming is an important variable that is integral to a more sustainable healthcare system. A recent cross-sectional study by Patel evaluated 49,329 telemedicine visits by 23,228 patients to the Moffitt Cancer Cancer (MCC) from April 1, 2020 to June 30, 2021. Patients living within a driving distance of 60 minutes from the cancer center saved on average 19.8 kg $CO_2$ emissions per-visit due to telemedicine, and for patients who were driving for more than 60 minutes, 98.6 kg $CO_2$ emissions was saved per visit [56]. While their work and results align with our findings, the scales are different as the population density is approximately 18.5 times higher in Istanbul, Turkey (2,523 people/km$^2$ or 6,530 people/mi$^2$) compared to Florida, USA (136.4 people/km$^2$ or 353.4 people/mi$^2$), which could explain the longer travel distance and time per visit in Florida [57, 58].

Despite the extensive effectiveness research, cost and perceptions of telemedicine, there have been few contributions assessing the environmental impact, such as travel times [41, 59–61].

Telemedicine could help to contain the increasing healthcare costs in a sustainable way if used in appropriate settings and accepted by the society while mitigating the risk for health disparities. We therefore attempted to define the impact of the current healthcare delivery model of in-person visits to a busy healthcare system in Istanbul, Turkey, with the goal of developing a framework of a cost-conscious and sustainable model [62]. Based on our data, the average person visited the clinics from a distance of 47.6 ± 96.3 km and spent 67.2 ± 108.9 min travel time and associated $CO_2$ emission 4,998 (bus)/ 9,139 gr (car).

The loss of productivity also impacts the sustainability. Assuming that only 57.4% of the less than 60 year old patients were employed, the average wage loss was $4.20/patient ([two-way travel time + 10 min physician time (hr)]*$3.26). Our data suggests that using telemedicine would lead to significant cost savings as well as a positive impact on the environment. In addition, the existence of a single payor system in Turkey brings advantages to test the telemedicine model on a larger scale.

Telemedicine has been practiced since early 1970s and its adoption accelerated with the Covid-19 pandemic [63, 64]. The reported advantages include lower healthcare costs, high patient satisfaction, improved access, decreased wait times and fewer missed appointments. The potential disadvantages include depersonalization of the clinician–patient relation and

concerns around quality of care along with the risk for perpetuating health disparities, the 'digital divide'. Thus, it is important to study and address the potential disadvantages of incorporating telemedicine to routine care including, such as access to digital infrastructure, education, and literacy [65–70]

In the case of Turkey, telemedicine acceptance was fast during the COVID-19 pandemic, yet the capacity for expansion, financing, policies, governance, and partnership are critical factors for any new system to be successful [71, 72]. The hypothetical benefits of adopting telemedicine presented in this work could inform and motivate future policies.

## 5. Limitations

While it is exciting and promising to use telemedicine in delivering care to patients, there are several limitations for our work.

First of all, we used data from a very specialized healthcare system serving patients with cardiovascular diseases and this may not be applicable to other healthcare systems.

Secondly, as for the travel time calculations, we relied on the Google Maps algorithms and used the after-hours travel, however the traffic patterns may change and negatively impact our time calculations and increase the time spent traveling. We defined these travel times as the "floor" times and these results should only be treated as the minimum possible travel time. In addition, the wait times during in-person visits are omitted from our calculations for the wage loss as we did not have the data for the wait times, and this could be a future direction of our work.

Finally, realistic estimation of work hours and wage loss requires access to more granular data about the employment conditions of patients, which we did not have in our study.

## 6. Conclusion

In this paper, we examined the potential impact of delivering telemedicine to patients by using a dataset that spans more than seven years from three cardio-vascular clinics in Istanbul, Turkey. We estimated the carbon footprint of traveling for in-person visits and the incurred wage loss due to time spent for these visits. Given the limitation of our study, further research is necessary to explore the application of telemedicine in Turkey in order to effectively decrease the burden on patients, environment, increase access, and prevent wage loss caused by unnecessary hospital visits. Telemedicine could also become a valuable tool for developing sustainable cities of the future.

## Author Contributions

**Conceptualization:** Figen Özen, A. Murat Kaynar.

**Data curation:** Alptug H. Kaynar, Melike Elif Teker Açıkel, A. Murat Kaynar.

**Formal analysis:** Figen Özen, Alptug H. Kaynar, Z. Dilsun Kaynar, A. Murat Kaynar.

**Writing – original draft:** A. Murat Kaynar.

**Writing – review & editing:** A. Kubilay Korkut, Melike Elif Teker Açıkel, Z. Dilsun Kaynar, A. Murat Kaynar.

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
