## [Decision Letter · Decision Letter 0]

11 Jun 2024

PONE-D-24-12793The role of telemedicine towards improved sustainability in health care and societal productivity in TurkeyPLOS ONE

Dear Dr. Kaynar,

Thank you for submitting your manuscript to PLOS ONE. After careful consideration, we feel that it has merit but does not fully meet PLOS ONE’s publication criteria as it currently stands. Therefore, we invite you to submit a revised version of the manuscript that addresses the points raised during the review process.

We look forward to receiving your revised manuscript.

Kind regards,

Boyen Huang, DDS, MHA, PhD

Academic Editor

PLOS ONE

Journal Requirements:

4. We note that Figure 4 in your submission contain map images which may be copyrighted. All PLOS content is published under the Creative Commons Attribution License (CC BY 4.0), which means that the manuscript, images, and Supporting Information files will be freely available online, and any third party is permitted to access, download, copy, distribute, and use these materials in any way, even commercially, with proper attribution. For these reasons, we cannot publish previously copyrighted maps or satellite images created using proprietary data, such as Google software (Google Maps, Street View, and Earth). For more information, see our copyright guidelines: http://journals.plos.org/plosone/s/licenses-and-copyright.

A. You may seek permission from the original copyright holder of Figure 4 to publish the content specifically under the CC BY 4.0 license.  

B. If you are unable to obtain permission from the original copyright holder to publish these figures under the CC BY 4.0 license or if the copyright holder’s requirements are incompatible with the CC BY 4.0 license, please either i) remove the figure or ii) supply a replacement figure that complies with the CC BY 4.0 license. Please check copyright information on all replacement figures and update the figure caption with source information. If applicable, please specify in the figure caption text when a figure is similar but not identical to the original image and is therefore for illustrative purposes only.

Reviewers' comments:

Reviewer's Responses to Questions

**Comments to the Author**

1. Is the manuscript technically sound, and do the data support the conclusions?

Reviewer #1: Yes

Reviewer #2: Yes

Reviewer #3: Partly

Reviewer #4: No

2. Has the statistical analysis been performed appropriately and rigorously? 

Reviewer #1: Yes

Reviewer #2: Yes

Reviewer #3: No

Reviewer #4: No

3. Have the authors made all data underlying the findings in their manuscript fully available?

Reviewer #1: No

Reviewer #2: Yes

Reviewer #3: Yes

Reviewer #4: No

4. Is the manuscript presented in an intelligible fashion and written in standard English?

Reviewer #1: Yes

Reviewer #2: Yes

Reviewer #3: No

Reviewer #4: No

5. Review Comments to the Author

Reviewer #1: Comments for the authors

In the manuscript “The role of telemedicine towards improved sustainability in health care and societal productivity in Turkey” the authors have calculated the average distance and time of patients from three clinics in Istanbul, from 2015 to 2023. In all, 45602 visits were analyzed. They found that patients traveled 23.82 ± 96.3 km to reach the clinics. The authors claim that 656,258 km would have been saved if all patients were to take the first visit in person followed by telemedicine visits. They further estimate that exploiting telemedicine could have saved approximately 30% carbon footprint and prevented approximately $878,000 wage loss.

The topic is important and timely. The manuscript is mostly well written, although it could be shortened quite a bit and better structured. The discussion should be more focused and to the point- especially the start. Figures 1-5 are unneccessary.

I find that the information in this manuscript is of scientific interest. However, the analyses, interpretations and conclusions are probably too positive with regard to telemedicine. The assumption that all visits after the first visit could be performed with telemedicine is highly problematic. Further, health-risks and quality are not discussed and the estimation of work-hours is not realistic- e.g. not all patients are employed. Thus, the manuscript does not comprehensively present a realistic scenario regarding cost/benefits, risks and actual implementation. This should receive more attention.

The major weakness concern the assertion of all visits post first visit can be done with telemedicine. This should at least be ameliorated, as this is directly relevant for the estimates

Reviewer #2: The paper is on an interesting topic, looking into the role of telemedicine and whether the use of this technology could help in saving time and reduce the carbon footprint. The authors could expand more on the background session. Methodology part is clear and well presented, as well as the discussion. It is good that you looking into the carbon footprint and the economic gains of using telemedicine, but it would be useful to look also in the quality of these consultations, the intangible qualities of telemedicine. I don’t know for example, if you are aware of the paper by Chonglin Sun et al. on Telemedicine Practice for Social and Environmental Sustainability and other works on temeledince and sustainable social development.

Reviewer #3: In the abstract the cost savings were for clinic visits yet it reads that cost savings may occur from hospital visits. How do you know that hospital visits may have occurred? The first two sentences of the abstract are illogical--COVID is over so how can telemedicine alleviate the current burden? The abstract has too many errors to note. The authors toggle between the terms telehealth and telemedicine--they are different and consistency is needed. What does recent mean? Two minutes ago, two decades ago? Asynchronous is not always pre-recorded information. Wage loss in confusing. You seem to be assuming everyone is working yet some are retired, on disability, etc. Why are the ages for males <60 and females <58? Is there a lower limit?

Reviewer #4: Thank you for the opportunity to read this paper the contribution of telemedicine on sustainability of health care. This paper aims to examine: i) the hypothetical impact of delivering telehealth care to patients in a busy tertiary cardiovascular clinic in Istanbul, Turkey; ii) potential environmental and societal ramifications of telemedicine. To address this issue, authors used demographics, health care costs, wages, productivity, and patient-specific data. The main goal is to develop a hypothetical telemedicine framework for the Turkish health care landscape using information related to the distance traveled and travel time to receive care using real-life location of the clinics and patients addresses.

Data from August 3, 2015 to January 25, 2023 resulting in 45,602 unique encounters with 448 unique diagnoses recorded for the patient encounters. Authors reported that 656,258 km would have been saved if all patients were to take the first visit in person followed by

telemedicine visits. Telemedicine could save approximately 30% carbon footprint and prevented approximately $ 878,000 wage loss. They concluded that the application of telemedicine could ease the burden on patients, environment, increase access, and prevent the wage losses caused by unnecessary hospital visits.

Introduction: The introduction should serve as a comprehensive literature review, setting the stage for your study. Unfortunately, this section is currently a major weakness in your manuscript. To strengthen it,you could consider the following points:

1. Begin by presenting an overview of cardiovascular diseases (CVDs), including their prevalence, impact, and burden on public health globally, and then narrow down to the context of Turkey. Use up-to-date statistics and references to highlight the significance of the problem.

2. Emphasize the need for innovative interventions to address the burden of CVDs. Introduce digital technologies, particularly telemedicine, as a potential solution.

3. Provide a clear definition of telemedicine and discuss its potential benefits and challenges, including barriers to adoption and existing knowledge gaps.

4. Present the relevance and potential value of a hypothetical telemedicine framework in mitigating the burden of CVDs.

Materials and Methods: There are some ambiguities and a lack of structure in this section. For instance, I will encourage the authors to follow standard headings and subheadings such as: study design, population and settings; measurement outcomes, dependent and Independent variables, ypes of measurements, data analysis; Could you explain the term"declaration." ; Provide a detailed and systematic description of your methods to ensure reproducibility and clarity.

Results: The results section should be presented clearly and logically. Currently, it is challenging to understand the study's objectives and the significance of the findings. I will suggest to authors to re-organize their results in a logical manner, corresponding to your research questions or hypotheses, highlight the key findings and their relevance to the study's objectives.

Limitations: The limitations section needs to be more precise. Specifically, please avoid overgeneralizations, such as comparing a single department with an entire healthcare system. Authors should discuss specific limitations related to your study design, data collection, analysis, and interpretation, as well as their potential impact on your conclusion. By addressing these points, you can significantly improve the quality and impact of your manuscript. I encourage you to thoroughly revise each section to ensure clarity, coherence, and rigor.

6. PLOS authors have the option to publish the peer review history of their article (what does this mean?). If published, this will include your full peer review and any attached files.

Reviewer #1: No

Reviewer #2: No

Reviewer #3: No

Reviewer #4: No

---

## [Author Response · Author response to Decision Letter 0]

7 Aug 2024

Critique: We note that you have indicated that there are restrictions to data sharing for this study. For studies involving human research participant data or other sensitive data, we encourage authors to share de-identified or anonymized data. However, when data cannot be publicly shared for ethical reasons, we allow authors to make their data sets available upon request. 

Response: We changed the data sharing to anonymized data sets available upon request.

Critique: Please include your full ethics statement in the ‘Methods’ section of your manuscript file. In your statement, please include the full name of the IRB or ethics committee who approved or waived your study, as well as whether or not you obtained informed written or verbal consent. If consent was waived for your study, please include this information in your statement as well.

Response: We updated the Materials and Methods section to indicate that consent was waived for this study and information on the ethics committee is also given in that section. 

Critique: We note that Figure 4 in your submission contain map images which may be copyrighted. All PLOS content is published under the Creative Commons Attribution License (CC BY 4.0), which means that the manuscript, images, and Supporting Information files will be freely available online, and any third party is permitted to access, download, copy, distribute, and use these materials in any way, even commercially, with proper attribution. For these reasons, we cannot publish previously copyrighted maps or satellite images created using proprietary data, such as Google software (Google Maps, Street View, and Earth). For more information, see our copyright guidelines: http://journals.plos.org/plosone/s/licenses-and-copyright.

Response: We removed Figures 1-3 per reviewer suggestions and renamed Figure 4 as Figure 1. Current Figure 1 is intended to convey the same information as the old Figure 4 and it has been obtained from https://earthexplorer.usgs.gov, which is a public domain.

Group comments 

We thank the reviewers for their constructive feedback, which led us to rewrite the introduction and make several key changes to make the thesis of our paper clearer. We state below how our revision addresses reviewer comments in an itemized fashion.

Reviewer #1: 

Critique: The topic is important and timely. The manuscript is mostly well written, although it could be shortened quite a bit and better structured. The discussion should be more focused and to the point- especially the start. Figures 1-5 are unnecessary.

Response: Thank you for the input. We significantly edited the manuscript, by making its focus clearer and removing Figures 1-3.

Critique: …, the analyses, interpretations and conclusions are probably too positive with regard to telemedicine. The assumption that all visits after the first visit could be performed with telemedicine is highly problematic. Further, health-risks and quality are not discussed and the estimation of work-hours is not realistic- e.g. not all patients are employed. Thus, the manuscript does not comprehensively present a realistic scenario regarding cost/benefits, risks and actual implementation. This should receive more attention.

Response: We rewrote several sections of the manuscript making a clearer distinction between our conclusions for the specific context of our study (specific CVD diagnoses) based on our dataset and what we think about the potential benefits of telemedicine in general. We also expanded the Discussion and Limitations sections to include potential problems with telemedicine in more detail.

We acknowledge the critique about work-hours assumption applying to all patients. Therefore, we incorporated a new reference to a manuscript by Stratmann and modified our claim to say “…. -unless the visit leads to gain in health and subsequent productivity [62–67].” (Line 274). Moreover, we added discussion of this issue in the Limitations section.

Critique: The major weakness concern the assertion of all visits post first visit can be done with telemedicine. This should at least be ameliorated, as this is directly relevant for the estimates.

Response: We respectfully believe that the top 5% diagnoses such as venous insufficiency used to build our model can safely be managed through telemedicine after the first in-person visit.

Reviewer #2: 

Critique: The paper is on an interesting topic, looking into the role of telemedicine and whether the use of this technology could help in saving time and reduce the carbon footprint. The authors could expand more on the background session. 

Response: We restructured and rewrote several sections, including more references as appropriate.

Critique: Methodology part is clear and well presented, as well as the discussion. It is good that you looking into the carbon footprint and the economic gains of using telemedicine, but it would be useful to look also in the quality of these consultations, the intangible qualities of telemedicine. I don’t know for example, if you are aware of the paper by Chonglin Sun et al. on Telemedicine Practice for Social and Environmental Sustainability and other works on telemedicine and sustainable social development.

Response: As we did not have the actual telemedicine consultations, we did not have the actual telemedicine data, however we made further discussion points based on related work, including the work by Dr. Sun et al. 

Reviewer #3: 

Critique: In the abstract the cost savings were for clinic visits yet it reads that cost savings may occur from hospital visits. How do you know that hospital visits may have occurred? 

Response: In this manuscript, we used the words clinic and hospital interchangeably, meaning physical locations to which patients travel for receiving medical care.

Critique: The first two sentences of the abstract are illogical--COVID is over so how can telemedicine alleviate the current burden? 

Response: We removed the COVID statement from the abstract. 

Critique: The abstract has too many errors to note. The authors toggle between the terms telehealth and telemedicine--they are different and consistency is needed. What does recent mean? Two minutes ago, two decades ago? Asynchronous is not always pre-recorded information. Wage loss in confusing. You seem to be assuming everyone is working yet some are retired, on disability, etc. Why are the ages for males <60 and females <58? Is there a lower limit?

Response: We carefully revised the manuscript, making consistent references to telemedicine, time frames, synchronous vs. asynchronous.

As a clarification to the reviewer, Turkey has a retirement age of 60 for males and 58 for females and that was the reason for our earlier assumptions. This was stated in the original manuscript as well as the revised one in the section “Economic impact of traveling to healthcare facilities” starting on Line 136.

Reviewer #4: 

Thank you for the opportunity to read this paper the contribution of telemedicine on sustainability of health care. This paper aims to examine: i) the hypothetical impact of delivering telehealth care to patients in a busy tertiary cardiovascular clinic in Istanbul, Turkey; ii) potential environmental and societal ramifications of telemedicine. To address this issue, authors used demographics, health care costs, wages, productivity, and patient-specific data. The main goal is to develop a hypothetical telemedicine framework for the Turkish health care landscape using information related to the distance traveled and travel time to receive care using real-life location of the clinics and patients addresses.

Data from August 3, 2015 to January 25, 2023 resulting in 45,602 unique encounters with 448 unique diagnoses recorded for the patient encounters. Authors reported that 656,258 km would have been saved if all patients were to take the first visit in person followed by

telemedicine visits. Telemedicine could save approximately 30% carbon footprint and prevented approximately $ 878,000 wage loss. They concluded that the application of telemedicine could ease the burden on patients, environment, increase access, and prevent the wage losses caused by unnecessary hospital visits.

Critique: The introduction should serve as a comprehensive literature review, setting the stage for your study. Unfortunately, this section is currently a major weakness in your manuscript. To strengthen it, you could consider the following points:

1. Begin by presenting an overview of cardiovascular diseases (CVDs), including their prevalence, impact, and burden on public health globally, and then narrow down to the context of Turkey. Use up-to-date statistics and references to highlight the significance of the problem.

2. Emphasize the need for innovative interventions to address the burden of CVDs. Introduce digital technologies, particularly telemedicine, as a potential solution.

3. Provide a clear definition of telemedicine and discuss its potential benefits and challenges, including barriers to adoption and existing knowledge gaps.

4. Present the relevance and potential value of a hypothetical telemedicine framework in mitigating the burden of CVDs.

Response: We sincerely appreciate your constructive criticism. We restructured the manuscript and rewrote the introduction and revised the subsequent sections with your guidance. 

Critique: Materials and Methods: There are some ambiguities and a lack of structure in this section. For instance, I will encourage the authors to follow standard headings and subheadings such as: study design, population and settings; measurement outcomes, dependent and independent variables, types of measurements, data analysis; Could you explain the term “declaration." 

Provide a detailed and systematic description of your methods to ensure reproducibility and clarity.

Response: We restructured and edited the materials and methods sections.

Critique: Results: The results section should be presented clearly and logically. Currently, it is challenging to understand the study's objectives and the significance of the findings. I will suggest to authors to re-organize their results in a logical manner, corresponding to your research questions or hypotheses, highlight the key findings and their relevance to the study's objectives.

Response: We restructured the results section to follow the logical flow of data and reflect the order in the materials and methods section.

Critique: Limitations: The limitations section needs to be more precise. Specifically, please avoid overgeneralizations, such as comparing a single department with an entire healthcare system. Authors should discuss specific limitations related to your study design, data collection, analysis, and interpretation, as well as their potential impact on your conclusion. By addressing these points, you can significantly improve the quality and impact of your manuscript. I encourage you to thoroughly revise each section to ensure clarity, coherence, and rigor.

Response: We made major changes in our manuscript along these lines. We pay attention to making a clearer distinction between conclusions for the specific context of our study, based on our data, and what we think about the potential benefits of telemedicine in general.

---

## [Decision Letter · Decision Letter 1]

2 Oct 2024

PONE-D-24-12793R1The role of telemedicine towards improved sustainability in healthcare and societal productivity in TurkeyPLOS ONE

Dear Dr. Kaynar,

Thank you for submitting your manuscript to PLOS ONE. After careful consideration, we feel that it has merit but does not fully meet PLOS ONE’s publication criteria as it currently stands. Therefore, we invite you to submit a revised version of the manuscript that addresses the points raised during the review process.

**Please be advised to address the reviewers' comments thoroughly, particularly on (1) justification of the participants' employment status, (2) recognition of the global history of telemedicine development and evolution, (3) the length of the discussion section, and (4) appropriate references.**

We look forward to receiving your revised manuscript.

Kind regards,

Boyen Huang, DDS, MHA, PhD

Academic Editor

PLOS ONE

Reviewers' comments:

Reviewer's Responses to Questions

**Comments to the Author**

1. If the authors have adequately addressed your comments raised in a previous round of review and you feel that this manuscript is now acceptable for publication, you may indicate that here to bypass the “Comments to the Author” section, enter your conflict of interest statement in the “Confidential to Editor” section, and submit your "Accept" recommendation.

Reviewer #1: (No Response)

Reviewer #3: All comments have been addressed

Reviewer #4: (No Response)

2. Is the manuscript technically sound, and do the data support the conclusions?

Reviewer #1: Partly

Reviewer #3: Yes

Reviewer #4: Partly

3. Has the statistical analysis been performed appropriately and rigorously? 

Reviewer #1: No

Reviewer #3: Yes

Reviewer #4: Yes

4. Have the authors made all data underlying the findings in their manuscript fully available?

Reviewer #1: No

Reviewer #3: Yes

Reviewer #4: No

5. Is the manuscript presented in an intelligible fashion and written in standard English?

Reviewer #1: Yes

Reviewer #3: Yes

Reviewer #4: No

6. Review Comments to the Author

**Reviewer #1:** In the R1 version of the manuscript " The role of telemedicine towards improved sustainability in healthcare and societal productivity in Turkey", the authors have submitted a revised version and responded to reviewer comments.

While the manuscript is improved, the authors have not remedied my main objection about the assumption that all patients below pension-age are employed. Do everybody below 58 years of age in Turkey have full-time jobs? I guess not and this must be described and modelled.

**Reviewer #3:** The authors state, "Telemedicine is a recent development within healthcare." What do you mean by recent? Telemedicine has been around for decades. Figure 1 is difficult to read. May want to telehealth as a key word.

**Reviewer #4: **Authors addressed many comments, and the paper has been improved. However, some points still require clarification. My main comments are on the objective/methods and discussion sections. I have minor comments on the limitation and conclusion sections.

Objective: Our study focuses on patients with CVDs and the potential savings afforded by telemedicine in travel distance and time, and lost wages as delayed access to acute as well as chronic care is associated with increased mortality.

• The following sentences should be moved to the method section

We also calculated 71 the hypothetically associated savings in travel distance and time for hospital visits enabled by 72 synchronous video conferences. The busy cardiology and cardio-vascular surgery clinics in 73 Istanbul used in our study can also be viewed as a model for studying the potential positive patient 74 outcomes, environmental, economic, and societal ramifications of deploying telemedicine. 75 Moreover, the location of these clinics can demonstrate the benefits of telemedicine in settings 76 where specialized care is concentrated in certain clinics in a populous city only accessible via often 77 traffic-laden travel routes. 78 We obtained datasets for demographics, healthcare costs, wages, productivity, and data of 79 patients of the above-mentioned clinics to develop a hypothetical telemedicine framework. 80 Specifically, we obtained the distance and time expended to receive care at the clinics by using the 81 addresses of the clinics and patients seeking care. The calculated distances and travel times allowed 82 us to calculate the carbon footprint and wage losses associated with traveling to seek in-person 83 care in lieu of telemedicine appointments. 84 In the case of Turkey, telemedicine acceptance was fast during the COVID-19 pandemic, yet 85 the capacity for expansion, financing, policies, governance, and partnership are critical factors for 86 any new system to be successful [22,23]. The hypothetical benefits of adopting telemedicine 87 presented in this work could inform and motivate future policies in this space.

Could they add references here

• The carbon footprint of traveling by bus is 105 grams per kilometer and the carbon footprint of 165 traveling by car is 192 grams per kilometer.

• Ref 30 is unclear, please Check for your bibliography

• As the work by Sun et al. (Year) (ref) suggests, telemedicine has the potential to alleviate some of the problems associated with the overburdened healthcare systems worldwide [38]

• I was not able to see these figures 7,8,9 in legends

Discussion

Discussion is very long and the authors are not focussing on their results.

They could start their discussion at line 335. I invite them to emphasize on the results and discuss them with the literature.

Our study focuses on patients with CVDs and the potential savings afforded by telemedicine in travel distance and time, and lost wages as delayed access to acute as well as chronic care is associated with increased mortality.

This research aims about the savings afforded to telemedicine in travel distance and time, and lost wages….

"The authors reported the non-inferiority of telemedicine and its impact on quality-related outcomes. I would suggest they focus more closely on the specific goal of the study and avoid overgeneralization. The term 'quality of care' is broad, and the effects of telemedicine on overall quality remain unclear. They could narrow their discussion to a few specific components of quality of care rather than addressing it as a whole”

“Our findings are supported by the recent literature that has demonstrated the non-inferiority of 339 outcomes in telemedicine treatment compared with face-to-face treatment for patients with 340 peripheral arterial and venous diseases [45–47]. There are published and ongoing clinical trials for 341 personalized telemedicine around cardiovascular health [48]. A recent work compared the use of 342 telemedicine to in-person doctor visits among patients with congestive heart failure (CHF). Among 343 850 patients with CHF, a hybrid telemedicine care model was not inferior to in-person visits for 344 mortality [49]. In a trial among 1,119 patients with CHF, telemedicine-based management 345 emergency service for high-risk HF patients proved to be safe and reduced unplanned 346 hospitalizations [50]. Another trial showed decreased emergency medicine department admissions 347 among patients with CHF if they were followed up by a tele-pharmacist [51]. Based on the clinical 348 trials, there is an important and effective potential role for telemedicine in healthcare [52–54]”

Here , you could split in two sentences:

Delayed access to acute as well as chronic care is associated with increased mortality (._) and prior Some authors work showed the negative impact of long wait periods on CVD outcomes [39–41]

Limitations:

I recommended to keep 1 and 4. I am not sure th numbers 2 and 3 are relevant to your study.

Conclusion:

Your conclusion goes beyond the scope of your study. For instance, you could rewrite like this:

“This 440 way, telemedicine could become a valuable tool for sustainable cities of the future.”

Suggestion

In the limitation of your study, further research is necessary to explore …… in order to effectively promote telemedicine as a valuable tool for developing sustainable cities.

7. PLOS authors have the option to publish the peer review history of their article (what does this mean?). If published, this will include your full peer review and any attached files.

Reviewer #1: No

Reviewer #3: **Yes: **Lois Ritter

Reviewer #4: No

---

## [Author Response · Author response to Decision Letter 1]

2 Nov 2024

Prof. Huang:

Thank you for submitting your manuscript to PLOS ONE. After careful consideration, we feel that it has merit but does not fully meet PLOS ONE’s publication criteria as it currently stands. Therefore, we invite you to submit a revised version of the manuscript that addresses the points raised during the review process.

Please be advised to address the reviewers' comments thoroughly, particularly on (1) justification of the participants' employment status, (2) recognition of the global history of telemedicine development and evolution, (3) the length of the discussion section, and (4) appropriate references.

Response: Thank you, Prof. Huang, for these high-level critiques. In the rest of the document, we address each of them in detail. To summarize:

1- The employment status critique is very much appreciated for a better modeling of the wage loss. We used OECD data to refine our model.

2- We modified our writing such that we recognize the global history of telemedicine.

3- We shortened and focused the discussion.

4- We added appropriate references.

Comments to the Author

We thank the reviewers for their constructive feedback, which led us to rewrite the introduction and make several key changes to make the thesis of our paper clearer. We state below how our revision addresses reviewer comments in an itemized fashion.

Reviewer #1: 

Critique: In the R1 version of the manuscript " The role of telemedicine towards improved sustainability in healthcare and societal productivity in Turkey", the authors have submitted a revised version and responded to reviewer comments.

While the manuscript is improved, the authors have not remedied my main objection about the assumption that all patients below pension-age are employed. Does everybody below 58 years of age in Turkey have full-time jobs? I guess not and this must be described and modelled.

Response: We thank the reviewer for this very important critique. To estimate the number of people who were employed before the pension age, we used the latest employment rate data from the Organisation for Economic Co-operation and Development (OECD), available using the links below. This data shows the average employment to be 57.4% between 2015 through 2022 for the 15-64 age range regardless of the gender. 

https://www.oecd.org/en/publications/oecd-economic-surveys-turkiye-2023_864ab2ba-en.html

https://data-explorer.oecd.org/vis?lc=en&df[ds]=DisseminateArchiveDMZ&df[id]=DF_DP_LIVE&df[ag]=OECD&df[vs]=&av=true&pd=2015%2C2023&dq=TUR%2BOECD..15_64..A&to[TIME_PERIOD]=false&vw=tb

Reviewer #3: 

Critique: The authors state, "Telemedicine is a recent development within healthcare." What do you mean by recent? Telemedicine has been around for decades. Figure 1 is difficult to read. May want to telehealth as a key word.

Response: We thank the reviewer. We use a more careful language recognizing the fact that telemedicine has been practiced for decades while still making the point that its adoption has been accelerated by COVID-19 (page 18). 

We added “Telehealth” added as a key word (page 3).

Reviewer #4: 

Critiques: Authors addressed many comments, and the paper has been improved. However, some points still require clarification. My main comments are on the objective/methods and discussion sections. I have minor comments on the limitation and conclusion sections.

Critique:

Objective: Our study focuses on patients with CVDs and the potential savings afforded by telemedicine in travel distance and time, and lost wages as delayed access to acute as well as chronic care is associated with increased mortality.

• The following sentences should be moved to the method section

We also calculated 71 the hypothetically associated savings in travel distance and time for hospital visits enabled by 72 synchronous video conferences. The busy cardiology and cardio-vascular surgery clinics in 73 Istanbul used in our study can also be viewed as a model for studying the potential positive patient 74 outcomes, environmental, economic, and societal ramifications of deploying telemedicine. 75 Moreover, the location of these clinics can demonstrate the benefits of telemedicine in settings 76 where specialized care is concentrated in certain clinics in a populous city only accessible via often 77 traffic-laden travel routes. 78 We obtained datasets for demographics, healthcare costs, wages, productivity, and data of 79 patients of the above-mentioned clinics to develop a hypothetical telemedicine framework. 80 Specifically, we obtained the distance and time expended to receive care at the clinics by using the 81 addresses of the clinics and patients seeking care. The calculated distances and travel times allowed 82 us to calculate the carbon footprint and wage losses associated with traveling to seek in-person 83 care in lieu of telemedicine appointments. 84 In the case of Turkey, telemedicine acceptance was fast during the COVID-19 pandemic, yet 85 the capacity for expansion, financing, policies, governance, and partnership are critical factors for 86 any new system to be successful [22,23]. The hypothetical benefits of adopting telemedicine 87 presented in this work could inform and motivate future policies in this space.

Response: The sentences have been relocated.

Critique:

• Could they add references here

- ”The carbon footprint of traveling by bus is 105 grams per kilometer and the carbon footprint of 165 traveling by car is 192 grams per kilometer.”

- Ref 30 is unclear, please Check for your bibliography

- As the work by Sun et al. (Year) (ref) suggests, telemedicine has the potential to alleviate some of the problems associated with the overburdened healthcare systems worldwide [38]

• I was not able to see these figures 7,8,9 in legends

Response: Corrected and references added

The Ref 30 in R1 was about the impact of aging society on the health care expenditure. “The effect of population aging on health expenditure growth: a critical review. Eur J Ageing. 2013;10: 353–361. doi:10.1007/s10433-013-0280-x”

The following phrase was deleted: “As the work by Sun et al. (Year) (ref) suggests, telemedicine has the potential to alleviate some of the problems associated with the overburdened healthcare systems worldwide [38]”

We removed the Figures 7-9 and the associated legends.

Critique:

Discussion

Discussion is very long and the authors are not focusing on their results. They could start their discussion at line 335. I invite them to emphasize on the results and discuss them with the literature.

Our study focuses on patients with CVDs and the potential savings afforded by telemedicine in travel distance and time, and lost wages as delayed access to acute as well as chronic care is associated with increased mortality.

This research aims about the savings afforded to telemedicine in travel distance and time, and lost wages….

"The authors reported the non-inferiority of telemedicine and its impact on quality-related outcomes. I would suggest they focus more closely on the specific goal of the study and avoid overgeneralization. 

The term 'quality of care' is broad, and the effects of telemedicine on overall quality remain unclear. They could narrow their discussion to a few specific components of quality of care rather than addressing it as a whole”

“Our findings are supported by the recent literature that has demonstrated the non-inferiority of outcomes in telemedicine treatment compared with face-to-face treatment for patients with peripheral arterial and venous diseases [45–47]. There are published and ongoing clinical trials for personalized telemedicine around cardiovascular health [48]. A recent work compared the use of telemedicine to in-person doctor visits among patients with congestive heart failure (CHF). Among 850 patients with CHF, a hybrid telemedicine care model was not inferior to in-person visits for mortality [49]. In a trial among 1,119 patients with CHF, telemedicine-based management emergency service for high-risk HF patients proved to be safe and reduced unplanned hospitalizations [50]. Another trial showed decreased emergency medicine department admissions among patients with CHF if they were followed up by a tele-pharmacist [51]. Based on the clinical trials, there is an important and effective potential role for telemedicine in healthcare [52–54]”

Here, you could split in two sentences:

Delayed access to acute as well as chronic care is associated with increased mortality (._) and prior Some authors work showed the negative impact of long wait periods on CVD outcomes [39–41]

Limitations:

I recommended to keep 1 and 4. I am not sure th numbers 2 and 3 are relevant to your study.

Conclusion:

Your conclusion goes beyond the scope of your study. For instance, you could rewrite like this:

“This way, telemedicine could become a valuable tool for sustainable cities of the future.”

Suggestion

In the limitation of your study, further research is necessary to explore …… in order to effectively promote telemedicine as a valuable tool for developing sustainable cities.

Response: As can be observed in the revised manuscript, we followed the reviewer’s recommendations closely. We think that this led to a revised manuscript that is better organized, more focused and avoids over-generalizations.

7. PLOS authors have the option to publish the peer review history of their article (what does this mean?). If published, this will include your full peer review and any attached files.

Do you want your identity to be public for this peer review? For information about this choice, including consent withdrawal, please see our Privacy Policy.

Reviewer #1: No

Reviewer #3: Yes: Lois Ritter

Reviewer #4: No

---

## [Decision Letter · Decision Letter 2]

20 Nov 2024

The role of telemedicine towards improved sustainability in healthcare and societal productivity in Turkey

PONE-D-24-12793R2

Dear Dr. Kaynar,

We’re pleased to inform you that your manuscript has been judged scientifically suitable for publication and will be formally accepted for publication once it meets all outstanding technical requirements.

Kind regards,

Boyen Huang, DDS, MHA, PhD

Academic Editor

PLOS ONE

Additional Editor Comments (optional):

Reviewers' comments:

Reviewer's Responses to Questions

**Comments to the Author**

1. If the authors have adequately addressed your comments raised in a previous round of review and you feel that this manuscript is now acceptable for publication, you may indicate that here to bypass the “Comments to the Author” section, enter your conflict of interest statement in the “Confidential to Editor” section, and submit your "Accept" recommendation.

Reviewer #1: All comments have been addressed

Reviewer #4: All comments have been addressed

2. Is the manuscript technically sound, and do the data support the conclusions?

Reviewer #1: Yes

Reviewer #4: Yes

3. Has the statistical analysis been performed appropriately and rigorously? 

Reviewer #1: Yes

Reviewer #4: Yes

4. Have the authors made all data underlying the findings in their manuscript fully available?

Reviewer #1: No

Reviewer #4: Yes

5. Is the manuscript presented in an intelligible fashion and written in standard English?

Reviewer #1: Yes

Reviewer #4: Yes

6. Review Comments to the Author

Reviewer #1: My questions and objections has been resolved. I find the manuscript publishable and a valuable addition to the literature..

Reviewer #4: Authors have addressed all comments. The revisions have greatly enhanced clarity and overall coherence.

7. PLOS authors have the option to publish the peer review history of their article (what does this mean?). If published, this will include your full peer review and any attached files.

Reviewer #1: No

Reviewer #4: No

---

## [Editor Report · Acceptance letter]

25 Nov 2024

PONE-D-24-12793R2 

PLOS ONE

Dear Dr. Kaynar, 

I'm pleased to inform you that your manuscript has been deemed suitable for publication in PLOS ONE. Congratulations! Your manuscript is now being handed over to our production team.

Kind regards, 

on behalf of

Dr Boyen Huang 

Academic Editor

PLOS ONE